# The Effect of Apple and Pear Cultivars on In Vitro Fermentation with Human Faecal Microbiota

**DOI:** 10.3390/microorganisms13081870

**Published:** 2025-08-11

**Authors:** Anna M. E. Hoogeveen, Christine A. Butts, Caroline C. Kim, Carel M. H. Jobsis, Shanthi G. Parkar, Halina M. Stoklosinski, Kevin H. Sutton, Patricia Davis, Duncan I. Hedderley, Jason Johnston, Pramod K. Gopal

**Affiliations:** 1The New Zealand Institute for Plant and Food Research Limited (Plant & Food Research), Private Bag 11600, Palmerston North 4442, New Zealandcaroline.kim@plantandfood.co.nz (C.C.K.); carel.jobsis@plantandfood.co.nz (C.M.H.J.); shanthi.parkar@hotmail.com (S.G.P.); halina.stoklosinski@plantandfood.co.nz (H.M.S.); duncan.hedderley@plantandfood.co.nz (D.I.H.); pramod.gopal@plantandfood.co.nz (P.K.G.); 2The New Zealand Institute for Plant and Food Research Limited (Plant & Food Research), Private Bag 4707, Christchurch 8140, New Zealand; kevin.sutton@plantandfood.co.nz (K.H.S.); patricia.davis@plantandfood.co.nz (P.D.); 3The New Zealand Institute for Plant and Food Research Limited (Plant & Food Research), Private Bag 1401, Havelock North 4157, New Zealand; jason.johnston@plantandfood.co.nz; 4Riddet Institute, Massey University, Private Bag 11222, Palmerston North 4442, New Zealand

**Keywords:** apple, pear, pip fruit, gut microbiota, in vitro digestion, in vitro faecal fermentation, short-chain fatty acids, human health, dietary fibre, polyphenols

## Abstract

Apples and pears are among the most popular and frequently consumed fruits worldwide. The polyphenol and dietary fibre components of these fruits are known to influence the gut microbiota and the subsequent human health outcomes. This study investigated the effects of New Zealand grown apples and pears with differing polyphenol contents on the structure and function of the human gut microbiota. Five apple and two pear cultivars underwent in vitro human digestion and microbial fermentation. Samples taken at 0 and 18 h were analysed for changes in pH, microbial composition, and organic acid production. The change in pH after faecal fermentation was influenced by the type of fruit (apple or pear), with lower pH being observed in the apples. Significant apple or pear cultivar effects were observed for the gut microbiome and organic acid production. The apple cultivar ‘Golden Hornet’ produced the least butyrate and the greatest microbial alpha diversity, while the pear ‘PremP009’ showed greater butyrate production with increases in a butyrogenic species (*Acidaminococcus intestini*). Further studies are needed to investigate the effect of cultivar and type of fruit on nutrient absorption and microbial fermentation and the impact of these on human health.

## 1. Introduction

Apples (*Malus domestica*) and pears (*Pyrus* spp.) are among the most popular and frequently consumed fruits worldwide. Epidemiological studies have reported that pip fruit, when consumed frequently, may produce positive health outcomes including a reduced risk of developing chronic diseases such as cardiovascular disease, cancer, and diabetes [1,2,3,4]. Apple consumption has been shown to reduce the levels of blood glucose in overweight human adults [5] and to reduce cholesterol in the blood of healthy adults [6,7]. The polysaccharides (dietary fibre) and polyphenols found in the fruit may be responsible for these health benefits.

The dietary fibre contents of apples range from 1.4% to 2.7% (fresh weight), and 1.6 to 3.6% (fresh weight) for pears [8]. Pectin is the major soluble fibre in apples and pears, and the insoluble dietary fibres mainly consist of cellulose and hemicellulose [9]. Small quantities of fructo-oligosaccharides and galacto-oligosaccharides are also found in apples and pears [10]. These soluble and insoluble fibres are resistant to degradation by acid in the stomach and small intestinal enzymes and exert functional health benefits in the human gut. Fibre provides bulk and volume to ingested food, which affects gastric emptying, nutrient absorption, satiety and transit time, resulting in their reaching the large intestine relatively intact where they become substrates for microbial fermentation [11].

The total polyphenol concentrations in apples and pears are similar (6.4 and 6.6 µmol/g, respectively) [12]. They have distinct patterns with flavanols (catechin, epicatechin, procyanidins), flavonols (quercetin glycosides), phenolic acids (e.g., chlorogenic acid), dihydrochalcones (e.g., phloretin glycosides), and anthocyanins (e.g., cyanidin glycosides) predominant in apples [13,14,15]. While pears generally have a higher levels of chlorogenic acid and the addition of arbutin [16]. The proanthocyanidins in apple and pear are oligomers or polymers of flavan-3-ol units including (+)-catechin or (−)-epicatechin, with degrees of polymerisation (DP) of 2–4 units (oligomers: dimers, trimers, tetramers) and DP > 4 (polymers: pentamer, hexamer, etc.). The concentrations of these depend on the cultivar, growing region, and environmental conditions [17,18,19], maturity stage of the plant when harvested and shelf life. Polyphenols, such as procyanidins and most of the chlorogenic acids, escape absorption in the small intestine and accumulate in the large intestine, potentially serving as substrate for the gut microbiota [20,21]. Apples, especially when consumed with the peel, are a major contributor to the dietary polyphenolic intake of adult humans [22,23].

In the present study we used fresh apples and pears developed and grown in New Zealand to compare gut responses using in vitro digestion and absorption methods. This approach has been used previously to investigate the release of and metabolism of polyphenols in apples [24,25] and the prebiotic potential of kiwifruit [26,27,28]. The aim of this study was to determine the effects on the structure and function of human gut microbiota of New Zealand pip fruit cultivars (Rosaceae) with different polyphenol compositions using in vitro digestion and fermentation models.

## 2. Materials and Methods

### 2.1. Fruit

Five apple and two pear cultivars were obtained from The New Zealand Institute for Plant and Food Research Ltd. (PFR) orchards at Crosses Road, Havelock North, New Zealand (Table 1). The apple cultivars were selected based on their polyphenolic content, with ‘PremA129’ and ‘Scilate’ having relatively lower levels of polyphenols; ‘Hetlina’ and ‘Monty’s Surprise’ having a medium level of polyphenols; and ‘Golden Hornet’ having a higher level of polyphenols (unpublished PFR data). The fruits were obtained fresh and in a ‘ready to eat’ stage of ripeness. The fruits were prepared for digestion and fermentation by washing with tap water and drying using paper towels. The stems and cores were removed, and the remaining fruit tissue (including the skin) was sliced and coarsely pureed.

### 2.2. In Vitro Digestion and Fermentation

The apple and pear cultivars (Table 1) underwent in vitro digestion using the method developed by Monro et al. [29]. Sterile distilled water (heated to 121 °C for 15 min) was used as a ‘no dietary fibre’ control, and the apple and pear cultivars (Table 1) underwent in vitro digestion. The fibre control was inulin (Orafti^®^ Synergy1; BENEO-Orafti S.A., Oreye, Belgium), which is a mixture of long- and short-chain oligofructose and was not subjected to in vitro digestion. The fruits were pureed to a smooth paste, and 36 g (fresh weight) of each sample and the ‘water control’ were placed in 50 mL centrifuge tubes. The gastric digestion phase was initiated by adding 1.0 mL of 1.0 M hydrochloric acid to each tube to reduce the pH to 2.5 (gastric pH) and 1.0 mL of 10% (*w*/*v*) pepsin solution. Samples and control were placed in an agitating water bath for 30 min at 37 °C and with an agitation rate of 50 rpm.

After the gastric digestion phase, samples were neutralised by adding 2.0 mL of 1.0 M sodium bicarbonate and 5.0 mL of 0.2 M sodium maleate buffer. The small intestinal digestion phase was initiated by adding 5.0 mL of freshly prepared 2.5% porcine pancreatin solution to each sample or control, followed by 100 µL of amylglucosidase solution (234 U activity). The samples were then incubated in an agitating water bath (37 °C, 50 rpm for 120 min). The fruit digesta was stored at −80 °C until the in vitro fermentation.

To simulate the small intestinal absorption of small sugars and metabolites but retain the polyphenols, the samples were subjected to dialysis using a 10 kDa weight cutoff membrane against reverse osmosis (RO) water [27]. After 24 h, the dialysate was retained and loaded onto an Oasis™ solid-phase extraction (SPE) cartridge (20 cc, 1 g; Waters, MA, USA) that was conditioned with 5 mL methanol followed by 20 mL double-RO water using a vacuum manifold. The SPE cartridge was washed using 150 mL of double-RO water to remove salts and sugars. The polyphenolic compounds were eluted using 50 mL of methanol. The methanol was removed under reduced pressure on a rotary evaporator at 20 °C and then freeze-dried. The polyphenolic fractions were stored at −80 °C until re-addition to the in vitro fermentation stage.

Five healthy human adults who had not taken antibiotics in the last three months provided faecal samples for this study. The study was approved by the Central Health and Disabilities Ethics Committee in New Zealand under protocol 13/CEN/144. Within 30 min of collection, the faecal samples were mixed with anaerobic sterile chilled 0.01 M phosphate-buffered saline (PBS; pH 7.4; Sigma-Aldrich, Christchurch, New Zealand), with 10% *v*/*v* glycerol and 0.05% *w*/*v* cysteine added to create 20% *v*/*v* faecal slurry and made into 5 mL aliquots. The aliquots were frozen in liquid nitrogen and stored at −80 °C [27,28].

For the in vitro fermentation, a pooled faecal inoculum was prepared by mixing equal proportions of thawed faecal slurry (i.e., one aliquot from each donor) in an anaerobic chamber (Coy Laboratory Products Inc., Livonia, MI, USA) containing a mix of CO_2_/H_2_/N_2_ (5/5/90% *v*/*v*). Each dried phytochemical fraction was resuspended into the respective digesta. The digesta or inulin controls were then added to a sterile carbohydrate-free basal medium (pH 6.8) [27] and inoculated with pooled faecal inoculum at a final concentration of 1% (*v*/*v*) faeces. The fermentation was performed anaerobically at 37 °C for 18 h while shaking at 70 rpm. The final concentration was 25 g/100 mL of fruit substrate digesta or 1 g/100 mL of inulin. To monitor the fermentation processes, samples (two 1 mL aliquots) were taken immediately after inoculation (i.e., 0 h samples) and 18 h after inoculation. One aliquot was directly stored at −80 °C, while the other aliquot was centrifuged at 16,000× *g*, at 4 °C for 5 min, and the supernatant and pellet were stored separately at −80 °C.

### 2.3. Organic Acid Analysis

The supernatants collected after 18 h of in vitro fermentation were analysed for various organic acids (OAs) by gas chromatography. The OAs included formic, acetic, propionic, butyric, isobutyric, isovaleric, hexanoaic, heptanoic, lactic and succinic. The supernatant (100 µL) was diluted with 400 µL of 0.01 M phosphate-buffered saline (PBS) containing 2-ethylbutyric acid as an internal standard. The final concentration of 2-ethylbutyric acid was 5 mM. Standards were prepared containing 2-ethylbutyric acid (5 mM). Then, 250 µL of concentrated hydrochloric acid and 1000 µL diethyl ether were added to the samples and standards, after which they were centrifuged at 10,000× *g* and 4 °C for 5 min. The diethyl ether phase (100 µL) was derivatized using 20 µL of *N-tert*-butyldimethylsilyl-*N*-methyltrifluoroacetamide with 1% tert-butyldimethylchlorosilane (Sigma-Aldrich) in a water bath at 80 °C for 20 min. The samples and standards were held for 48 h at room temperature to allow complete derivatisation of lactate and succinate.

The analysis of the samples and standards was performed using a gas chromatograph (GC-2010, Shimadzu, Kyoto, Japan) fitted with a Restek column (SH-Rtx-1; 30 m × 0.25 mm ID × 0.25 μm; Shimadzu) and a flame ionisation detector. Helium was used as a carrier gas, with a total flow rate of 21.2 mL/min and a pressure of 131.2 kPa. Nitrogen was used as a make-up gas. The temperature programme began at 70 °C, increasing to 115 °C at 6 °C/min, with a final increase to 300 °C at 60 °C/min, held for 3 min. The flow control mode was set to linear velocity: 37.5 cm/s. The injector temperature was 260 °C, and the detector temperature was 310 °C. Samples were injected (1 µL) with a split ratio of 10:1. The GC instrument was controlled, and data were processed using Shimadzu GC Workstation LabSolutions Version 5.96. The acquired data were expressed as µmol organic acid/mL fermenta. The amounts of iso-butyric, iso-valeric, valeric, hexanoic, and heptanoic acid detected in the samples were negligible (i.e., below the detection limit), and therefore not reported here. The total organic acid production was defined as the sum of formic, acetic, propionic, butyric, lactic, and succinic acid concentrations.

### 2.4. Microbiota Analysis

Samples (1 mL) were centrifuged at 16,000× *g* for 5 min at 4 °C. DNA was extracted from the pellet using the DNeasy PowerLyzer PowerSoil Kit (QIAGEN Pty Ltd., Clayton, Victoria, Australia) following the manufacturer’s instructions. The homogenization was as follows: the samples were first heated at 70 °C for 10 min, then cooled on ice for 5 min, and subsequently homogenised in a bead beater (Fastprep-24 5G; MP Biomedicals, Irvine, CA, USA) at 4.0 m/s for 45 s. The quantity and purity of the extracted DNA were measured using a QIAxpert spectrophotometer (QIAGEN Pty Ltd.). The extracted DNA was then sent to the Massey Genome Service (Palmerston North, New Zealand) for dual-indexing sequencing on the Illumina^®^ MiSeq Sequencing platform to determine the taxonomic composition. Before the sequencing process, a PCR was conducted using Invitrogen AccuPrime™ Pfx SuperMix (17 µL; Invitrogen, Waltham, MA, USA), 10 µM 16SR_V4 Primer (1 µL), 10 µM 16SF_V3 Primer (1 µL) and 1 µL normalised DNA sample (5 ng/µL) to amplify variable regions V3–V4 of the 16S rRNA gene of the bacterial DNA. The primers used were 16SF_V3 (5′-AATGATACGGCGACCACCGAGATCTACAC-barcode-TATGGTAATTGGCCTACGGGAGGCAGCAG-3′) and 16SR_V4 (5′-CAAGCAGAAGACGGCATACGAGAT-barcode-AGTCAGTCAGCCGGACTACHVGGGTWTCTAAT-3′) [30], which also contained adaptors for the sequencing. The PCR began with a denaturation step at 95 °C for 2 min, followed by 30 cycles of 95 °C for 20 s, 55 °C for 15 s, 72 °C for 5 min, and then a final extension step at 72 °C for 10 min, followed by cooling at 4 °C. Subsequently, the PCR product was cleaned, normalised, and pooled using the Invitrogen SequalPrep™ Normalization Plate Kit (ThermoFisher Scientific, Waltham, MA, USA). The concentration of the library was determined using a Qubit DNA High-Sensitivity assay, and the library sizing was conducted using a Bioanalyzer DNA High-Sensitivity assay (Agilent Technologies, Santa Clara, CA, USA). The amplicons were pooled in equal molarity and 16S rRNA gene sequencing was performed on a 2 × 250 base paired-end run using the MiSeq Sequencer (Illumina, San Diego, CA, USA).

The Illumina amplicon sequences were analysed using Quantitative Insights Into Microbial Ecology 2 (QIIME 2; version 2023.7) [31]. The sequences were first quality-checked, then de-multiplexed, followed by a step in which sequencing errors and chimaeras were filtered by trimming using the DADA2 package [32] (trimmed by removing 5 bases from the sequences and truncating at base number 240). The output had a total of 1,073,160 frequencies and 873 features over 24 samples. The alpha diversity (Shannon diversity) and beta diversity (Bray–Curtis and weighted UniFrac distance) were determined using these outputs with a sampling depth set at 26,000 sequences per sample. The taxonomic alignment was performed using the Greengenes2 database (version 2022.10) [33] with 99% sequence similarity.

### 2.5. Data Analysis

The effects of substrate on pH value, total and individual organic acid productions, and Shannon diversity index were tested using a one-way analysis of variance (ANOVA). The same analysis was applied to test the effect of the type of pip fruit (i.e., apple or pear). The means were compared using the adjusted Tukey–Kramer test and were considered statistically different if *p* ≤ 0.05. The separation of the sample groupings after performing the Bray–Curtis and weighted UniFrac distance analyses was tested using a permutational analysis of variance (PERMANOVA). Taxa were analysed for differential abundance using the Analysis of Compositions of Microbiomes with Bias Correction (ANCOM-BC) [34] Plugin in QIIME2.

## 3. Results

### 3.1. pH Changes After In Vitro Human Faecal Fermentation

The pH after 18 h of in vitro fermentation was significantly affected (*p* ≤ 0.05) by the cultivar type of apple and pear substrates (Figure 1). For example, the pH of 18 h ‘Golden Hornet’ fermenta was 0.4 and 0.5 units higher (*p* ≤ 0.05), compared to that of ‘PremA129’ and ‘Scilate’, respectively. In general, the apple substrates (4.26 ± 0.01; mean ± SE) resulted in a lower (*p* ≤ 0.05) pH compared to the pear substrates (4.33 ± 0.01; mean ± SE).

### 3.2. Production of Organic Acids During In Vitro Faecal Fermentation

The pip fruit cultivar had a significant effect on the total organic acid production, as well as the production of individual organic acids after 18 h of in vitro human faecal fermentation (*p* ≤ 0.05; Figure 2 and Table 2). Organic acid production was highest for the ‘PremP009’ pear for formic, butyric and propionic acids, and lowest for succinic acid. The ‘PremA129’ and ‘Scilate’ apples produced the highest concentrations of acetic, lactic and succinic acids. However, butyric acid production was lowest for ‘PremA129’ and ‘Monty’s Surprise’ and highest for ‘PremP009’ and ‘Golden Hornet’ apples.

The type of pip fruit (i.e., apple or pear) had no effect on total organic acids but there were significant differences (*p* ≤ 0.05) for some of the individual organic acids (Table 3). Pears produced more formic, butyric, and propionic acids than apples, while apples produced more succinic acid than the pear samples.

### 3.3. Microbial Composition After In Vitro Faecal Fermentation

The α-diversity, based on the Shannon diversity index, of the microbial community after 18 h of in vitro fermentation was affected by the substrate (*p* ≤ 0.05; Figure 3). For example, with a Shannon diversity index of 5.94 ± 0.03 (mean ± SE), the ‘Golden Hornet’ substrate yielded the greatest α-diversity. In contrast, the ‘Pre-mA129’ and ‘Scilate’ substrates had the smallest α-diversity with 5.39 ± 0.04 and 5.43 ± 0.02, respectively. The β-diversity (Bray–Curtis dissimilarity and weighted UniFrac distance) of the microbial composition after 18 h of in vitro fermentation showed significant separation of the grouping of the different substrates (*p* ≤ 0.05), supported by the PERMANOVA test (Figure 4).

The relative abundance of the bacterial phyla (with at least 0.1% relative abundance in one sample) and the twenty most abundant species identified after 18 h of in vitro fermentation of various apple and pear substrates using a pooled human faecal inoculum are shown in Figure 5 and Figure 6, respectively. Based on analysis of compositions of microbiomes with bias correction (ANCOM-BC) analyses, 59 differentially abundant species existed between the various substrates after 18 h of in vitro fermentation compared to the 0 h samples (Figure 7). The ‘PremP009’ pear fermentation substrate produced the highest number of species with an increased (q ≤ 0.05) proportion of 34 different species (most notably *Akkermansia muciniphila*_D_776786, log-fold change (LFC) = 1.0; and *Bacteroides_H salyersiae*, LFC = 2.0) and a decreased (q ≤ 0.05) proportion of nine species. The ‘Monty’s Surprise’ substrate affected the differential abundance of the least number of species; enriched (q ≤ 0.05) the proportion of seventeen species and decreased (q ≤ 0.05) the proportion of four species. Overall, nine bacterial species were enriched (q ≤ 0.05) in all substrates after 18 h of in vitro fermentation, including *Bifidobacterium adolescentis* (LFC = 1.5 to 2.8) and *Sutterella wadsworthensis*_A_565807 (LFC = 3.4 to 4.3).

The effect of the type of pip fruit (apple and pear) on microbial diversity and composition was also investigated. Even though the Shannon diversity between the types of pip fruit (apple and pear) was similar (*p* = 0.561, Figure 8), type of fruit did influence (q ≤ 0.05) the differential abundance of eleven bacterial species after 18 h of in vitro fermentation (Figure 9), based on ANCOM-BC analysis. For example, *Allisonella histaminiformans* (LFC = 2.4) and *Bifidobacterium longum* (LFC = 1.0) were proportionally enriched (q ≤ 0.05) after fermenting the pear substrates. On the other hand, *Eubacterium plexicaudatum* (LFC = 2.7) and *Faecalibacillus intestinalis* (LFC = 1.1) were proportionally enriched (q ≤ 0.05) in the apple substrates.

## 4. Discussion

This study compared the in vitro fermentation properties of various types of pip fruit (i.e., apples and pears) and cultivars. We investigated how the two commonly consumed pip fruit affected the fermentation outcomes, including pH and organic acid production, and the bacterial community after 18 h of in vitro human faecal fermentation.

### 4.1. The Effect of Apple and Pear Substrates on Fermentation Outcomes After In Vitro Human Faecal Fermentation

We observed that the pH after fermentation of the apples and pears was inversely related to the total organic acid production. ‘PremA129’ and ‘Scilate’ apples produced the greatest amount of total organic acids, resulting in the lowest pH values after fermentation. In contrast, the ‘Golden Hornet’ apple had the highest pH value and lowest total organic acid production. This indicates that organic acid production is the primary driver responsible for the decrease in pH during fermentation. This agrees with a fermentation study using apple cultivars with different phenolic compound sizes that reported the suppression of short-chain fatty acid production with increased phenolic compound molecular size [15].

The greater total organic acid production observed for the ‘PremA129’ and ‘Scilate’ apple substrates can be largely attributed to the greater production of acetic, lactic and succinic acids. The relative proportion of acetic acid produced during the in vitro human faecal fermentation is consistent with previous findings [15]. However, the present study reported a higher proportion of propionic acid than butyric acid for all apple substrates (except ‘Golden Hornet’), which contradicts this earlier study [15]. It should be noted that the previous report used cider apple cultivars [15], which typically possess higher polyphenol content compared to the eating apple cultivars used in the present study (‘PremA129’, ‘Scilate’ and ‘Hetlina’). Both ‘Golden Hornet’ and ‘PremP009’ demonstrated significantly greater (*p* ≤ 0.05) production of butyric acid, which serves as an important energy source for epithelial cells [30] and has been suggested to reduce the risk of colorectal cancer [31,32,33]. While polyphenols in pip fruit may offer potential health benefits, it is worth noting that high polyphenolic content is often accompanied by a bitter taste, which can decrease consumer preference [34,35].

A unique aspect of the current study is that it included the investigation and comparison of two types of pip fruit, namely apples and pears. Previous research using in vitro human faecal fermentation of pip fruit has focused on apples [15,36,37]. We observed differences between the apple and pear cultivars whereby the pear varieties produced significantly greater (*p* ≤ 0.05) levels of formic, butyric and propionic acids compared to the apple varieties. This difference could be due to differences in the concentration and composition of microbiota-accessible carbohydrates, and the distinct polyphenol profiles of the apples and pear. Pears have 23% more total dietary fibre and 46% more insoluble dietary fibre than apples [8]. The various structural parameters of pectins, including their *α*-(1 → 4)-linked galacturonic acid linear chains, arabinose residues, conjugated protein content, and molecular weight affect their interaction with gut microbiota, the bacterial cross-feeding, and the microbial metabolic outcomes [38,39]. In addition, colonic pH regulates saccharolytic and proteolytic fermentation pathways in gut bacteria by altering bacterial composition, gene expression and enzymatic activity, thus affecting metabolic output such as the production of short-chain and branched chain fatty acids. Lower pH due to microbial fibre fermentation affects the generation of organic acids, predominantly the SCFA’s [40]. Further research is needed to unravel the complex interactions between food composition and structure, and the digestion and fermentation processes including the microbiota in the digestive system. In the current study, the analysis of microbial metabolites was limited to organic acids. However, it is known that the gut microbiota can utilise phenolic compounds from apples to produce metabolites like phenolic acids [15,41]. Additionally, galacturonic acid, derived from pectin (one of the major fibres in apples), was not assessed [42]. Therefore, it is recommended that future research explores a broader range of microbial metabolites.

### 4.2. Changes in Microbial Composition During In Vitro Faecal Fermentation Influenced by Apple and Pear Substrates

Both the α- and β-diversity measures were affected by the substrate, which may relate to the different fibre quantity and type as well as the bioactive composition (polyphenols) of the substrates. ‘Golden Hornet’, the apple substrate with the highest polyphenol concentration (internal PFR data), exhibited the greatest (*p* ≤ 0.05) Shannon diversity index. In contrast, ‘PremA129’ and ‘Scilate’, the apple substrates with the lowest polyphenol concentration (internal PFR data), clustered together in the Bray–Curtis dissimilarity analysis. The Bray–Curtis dissimilarity analysis also revealed clustering based on the type of pip fruit, possibly due to differences in soluble and insoluble fibre contents of apples and pears [8]. The Bray–Curtis dissimilarity measures the difference between microbial communities based on abundance counts, giving more weight to more abundant taxa [43]. In contrast, weighted UniFrac distances take into account the phylogenetic tree, showing the phylogenetic distance between community members [44]. Use of the weighted UniFrac metric resulted in less separation among the substrate clusters, indicating that the microbial communities were more closely related phylogenetically, which may be expected considering that a pooled faecal inoculum was used for the fermentation, and the substrates provided microbiota-accessible energy sources.

The current study found that certain bacterial species were enriched during in vitro fermentation of all substrates. For example, multiple species of *Bifidobacterium* were enriched (*q* ≤ 0.05), which aligns with prior studies that reported apples promoted the growth of *Bifidobacterium* in an in vitro fermentation model [37], and that pear pomace stimulated the growth of *Bifidobacterium* in a mouse model [45]. Also, *Bifidobacterium longum* has been described as capable of breaking down pectin and converting it into acetate and lactate [46]. *Bifidobacterium* might have been an important contributor to the acetate and lactate productions, especially in substrates showing the highest enrichment of these species (‘PremA129’ and ‘Scilate’; LFC = 2.4 and 2.7, respectively). Also, the comparison with the gold standard prebiotic inulin, indicates the strong prebiotic potential of these apple cultivars, as consumer-friendly whole foods, rather than as highly processed fibre supplements.

Some microbial species showed enrichment due to specific substrates. For example, *Acidaminococcus intestini*, a known butyrate-producer [47], was greatly enriched (*q* ≤ 0.05) in the ‘PremP009’ fermentation, likely contributing to the increased (*p* ≤ 0.05) butyrate production. Another species enriched (*q* ≤ 0.05) by the ‘PremP009’ substrate was *Akkermansia muciniphila*, whose increased abundance may have resulted from the higher concentrations of insoluble fibre in the pears compared to the apples [8]. Insoluble fibres have been shown to enhance the abundance of *Akkermansia muciniphila* in the faeces of rats fed a high-fat diet [48]. *Allisonella histaminiformans* was also found to be enriched (*q* ≤ 0.05) by the apples and pears in the present study, except for the ‘Scilate’ and ‘Hetlina’ apples. There is little published information on this bacterium other than it produces histamine and utilises histidine as its sole energy source [49]. While accumulation of histamine in the gut could potentially elevate the risk of gastrointestinal diseases, such as irritable bowel disease and colorectal cancer [50], the greater production of butyrate from these substrates may help mitigate these risks [31,32,33].

The apple and pear cultivars examined in this study were red-skinned, except for ‘Golden Hornet’. Research suggests that the peel contains the majority of polyphenolic compounds [51]. Although this study highlighted some differences between the substrates, a broader investigation that includes a wider range of varieties, such as green- and yellow-skinned, could provide a deeper understanding of how polyphenols affect gut microbiota and in vitro fermentation outcomes.

A limitation of this study was its exclusive focus on fermentation outcomes and changes in the microbial community. Previous reports indicate that polyphenols may undergo multiple changes during digestion [52]. For example, certain polyphenolic compounds can be broken down or degraded, resulting in the formation of additional isomers, such as those derived from chlorogenic acid [52]. Moreover, some polyphenolic compounds can be hydrolysed by the large intestinal microbiota. The apple polyphenol, chlorogenic acid is rapidly hydrolysed into caffeic acid, and then to smaller phenolic acids such as hydroxyphenyl acetic acids within 30 min of the initiation of microbial fermentation [53] which may then be absorbed by the epithelial cells [21]. For future research, it would be valuable to measure changes in polyphenolic compounds at various time points during in vitro digestion and fermentation. In the current study, polyphenolic compounds were separated from the dialysed material after in vitro digestion and added to the fermentation substrate under the assumption that these polyphenolic compounds would not be absorbed in the small intestine. However, other studies suggest that some polyphenolic compounds, and any microbial metabolites generated, may indeed be partially absorbed in the small intestine. For instance, one-third of chlorogenic acid and all of the caffeic acid were absorbed in the small intestine of human ileostomates [21]. A further limitation of this study was the comparison of 5 apple and 2 pear cultivars and the imbalance in numbers of apples compared to pears. These two types of fruit have differing fibre and polyphenol concentrations. Pears having 23% more total dietary fibre and 46% more insoluble dietary fibre than apple varieties.

## 5. Conclusions

In conclusion, the pip fruit type and cultivar affected the proliferation of microbial species and the production of organic acids during in vitro-simulated human fermentation. These effects may be attributed to their composition, especially their dietary fibre and polyphenolic contents, and the resulting differences in their digestion and fermentation by the gut bacteria. However, more research is warranted to understand these differences.

This study contributes to the understanding of the nutrient composition of different pip fruit cultivars that may contribute to the shaping of gut microbiota (i.e., stimulate the growth of beneficial microbes) and promote the production of beneficial organic acids (such as butyric acid). In turn, it will lead the way for targeted breeding to develop pip fruit cultivars with enhanced gut microbiome functionality and the potential to support beneficial health outcomes.

## Figures and Tables

**Figure 1 microorganisms-13-01870-f001:**
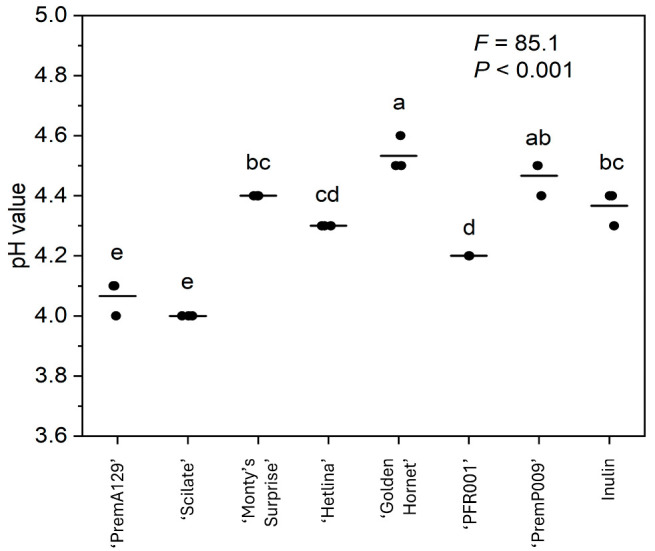
The pH at 18 h of in vitro fermentation of various apple and pear substrates (*n* = 3 per substrate) using a pooled human faecal inoculum. Data points represent individual replicates, and the line represents the mean per substrate. A one-way ANOVA model was used to assess the effect of substrate. Means with different letters differ across substrates (*p* ≤ 0.05).

**Figure 2 microorganisms-13-01870-f002:**
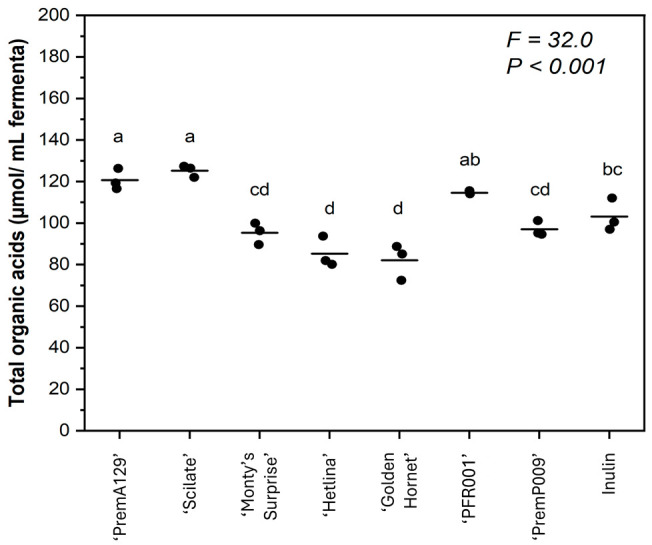
The total organic acid production after 18 h of in vitro fermentation of various apple and pear substrates (*n* = 3 per substrate) using a pooled human faecal inoculum. The total organic acid production was defined as the sum of formic, acetic, propionic, butyric, lactic, and succinic acid concentrations. Data points represent individual replicates, and the line represents the mean per substrate. A one-way ANOVA model was used to assess the effect of substrate. Means with different letters differ across substrates (*p* ≤ 0.05).

**Figure 3 microorganisms-13-01870-f003:**
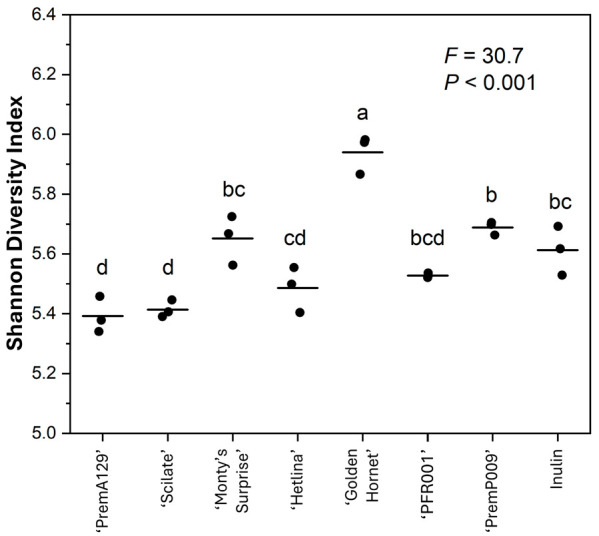
The Shannon diversity index after 18 h of in vitro fermentation of various apple and pear substrates (*n* = 3 per substrate) using a pooled human faecal inoculum. Data points represent individual replicates, and the line represents the mean per substrate. A one-way ANOVA model was used to assess the effect of substrate. Means with different letters differ across substrates (*p* ≤ 0.05).

**Figure 4 microorganisms-13-01870-f004:**
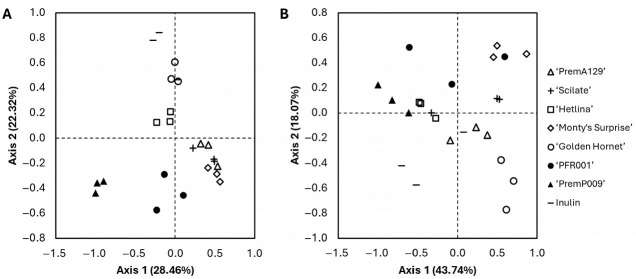
Principal Coordinate Analysis plots using the (**A**) Bray–Curtis dissimilarity and (**B**) weighted UniFrac distance of the microbial composition after 18 h of in vitro fermentation of various apple and pear substrates (*n* = 3 per substrate) using a pooled human faecal inoculum. Data points represent individual replicates. A PERMANOVA test (999 permutations) was used to assess the effect of substrate. The substrates clustering differed significantly for the Bray–Curtis dissimilarity (*p* = 0.001, *pseudo-F* = 10.0) and the weighted UniFrac distance (*p* = 0.001, *pseudo-F* = 7.5).

**Figure 5 microorganisms-13-01870-f005:**
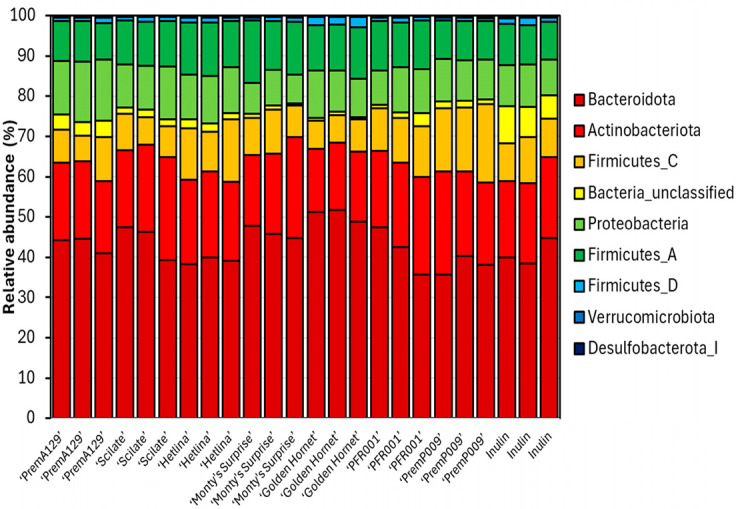
The relative abundance of phyla after 18 h of in vitro fermentation of various apple and pear substrates (*n* = 3 per substrate) using a pooled human faecal inoculum. Only phyla with at least 0.1% relative abundance in one sample were included.

**Figure 6 microorganisms-13-01870-f006:**
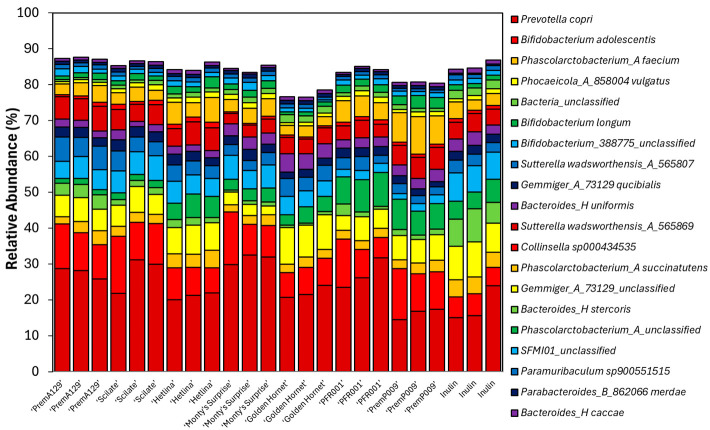
The relative abundance of the top 20 species after 18 h of in vitro fermentation of various apple and pear substrates (*n* = 3 per substrate) using a pooled human faecal inoculum.

**Figure 7 microorganisms-13-01870-f007:**
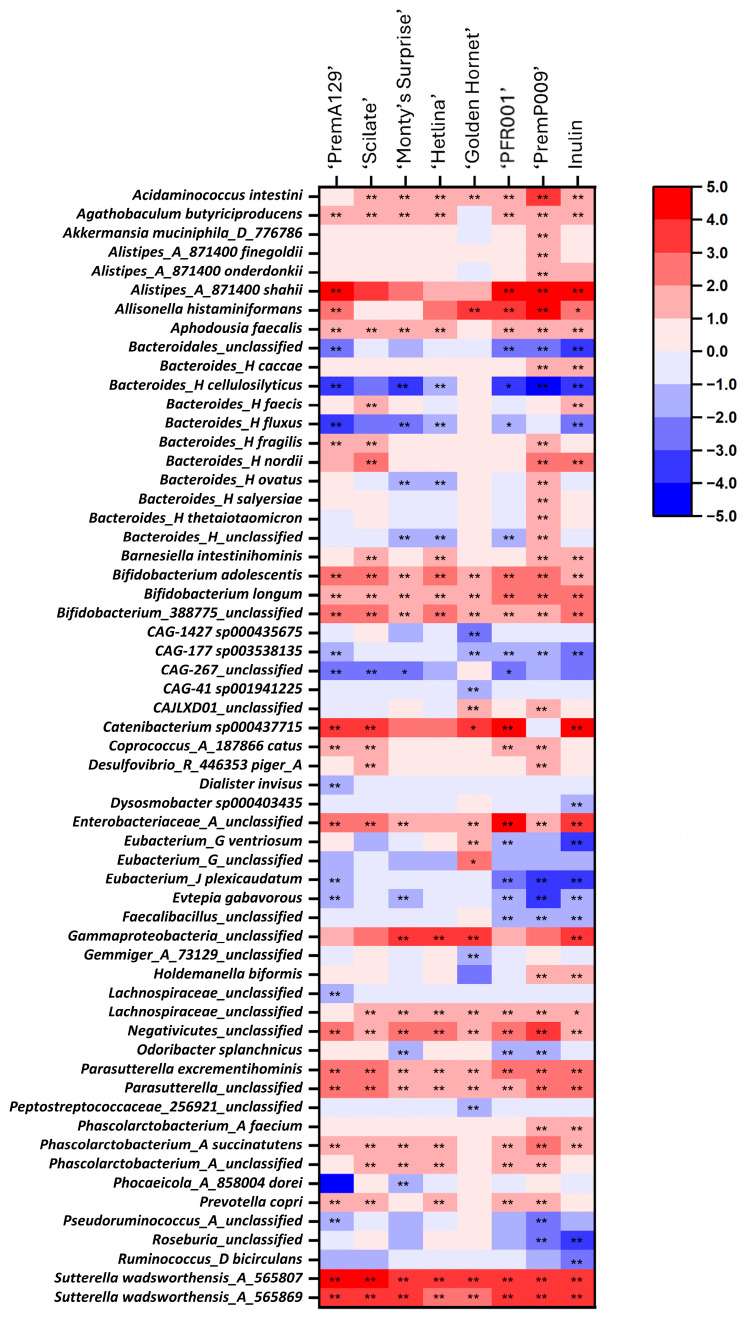
The differentially abundant bacterial species after 18 h of in vitro fermentation of the apple, pear, and inulin substrates with 0 h samples as the reference, based on ANCOM-BC analysis. The colour intensity indicates the size effects (LFC) of the species that were enriched (red) or depleted (blue). Only species with a relative abundance of ≥0.1%, that had an LFC of ≥1 and were significantly different (*q* ≤ 0.05) in at least one substrate were included in this figure. The FDR-adjusted (*q* ≤ 0.05) values are indicated by *, *q* ≤ 0.05; **, *q* ≤ 0.001. ANCOM-BC, analysis of compositions of microbiomes with bias correction; FDR, false discovery rate; LFC, log-fold change.

**Figure 8 microorganisms-13-01870-f008:**
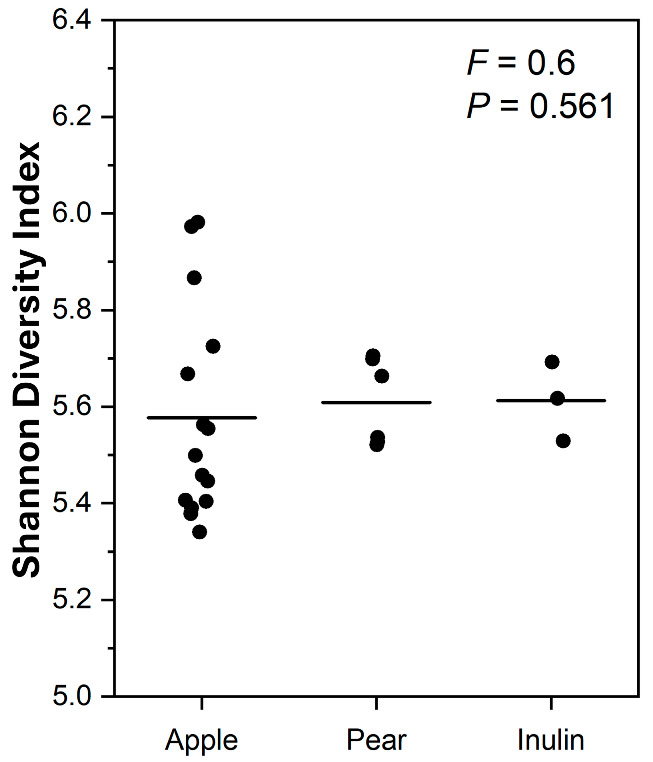
The Shannon diversity index after 18 h of in vitro fermentation of apple (*n* = 15), pear (*n* = 6), and inulin substrates (*n* = 3) using a pooled human faecal inoculum. Data points represent individual replicates, and the line represents the mean per substrate. A one-way ANOVA model was used to assess the effect of substrate. Means with different letters differ across substrates (*p* ≤ 0.05).

**Figure 9 microorganisms-13-01870-f009:**
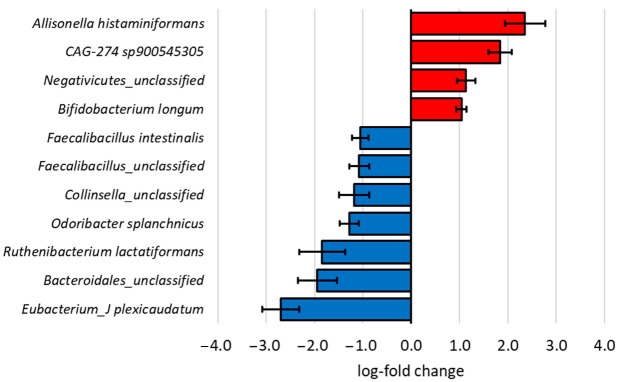
The differentially abundant bacterial species after 18 h of in vitro fermentation of pear substrates (*n* = 6) with apple substrates (*n* = 15) as the reference, based on ANCOM-BC analysis. Bars represent means ± SE. Only species with a log-fold change of at least 1 that were significantly (*q* ≤ 0.05) enriched (red) or depleted (blue) in proportion were included in this figure. ANCOM-BC, analysis of compositions of microbiomes with bias correction.

**Table 1 microorganisms-13-01870-t001:** Apple and pear cultivars included in this study.

Cultivars	Type of Pip Fruit	Total Polyphenol (μg g^−1^ Whole Fruit) ^1^
PremA129	apple	574
Scilate	apple	587
Monty’s Surprise	apple	772
Hetlina	apple	1336
Golden Hornet	apple	3152
PFR001	pear	NA
PremP009	pear	303

^1^ Average of samples obtained in the harvest years 2019–2023, determined by liquid chromatography-mass spectrometry (LC-MS); NA, not available.

**Table 2 microorganisms-13-01870-t002:** The individual organic acid productions after 18 h of in vitro fermentation per fruit substrates using a pooled human faecal inoculum ^1^.

Organic Acid	Substrate								*p* Value
	‘PremA129’	‘Scilate’	‘Monty’s Surprise’	‘Hetlina’	‘Golden Hornet’	‘PFR001’	‘PremP009’	INULIN	
	µmol/mL fermenta								
Formic	0.30 ± 0.00 ^d^	ND	1.27 ± 0.09 ^bc^	1.64 ± 0.06 ^b^	0.99 ± 0.01 ^c^	1.67 ± 0.09 ^b^	2.79 ± 0.21 ^a^	2.32 ± 0.04 ^a^	<0.001
Acetic	64.8 ± 1.15 ^a^	67.5 ± 0.52 ^a^	41.9 ± 0.88 ^d^	48.7 ± 1.29 ^c^	40.4 ± 1.13 ^d^	57.9 ± 0.92 ^b^	47.6 ± 0.85 ^c^	51.3 ± 1.03 ^c^	<0.001
Butyric	1.70 ± 0.10 ^e^	2.03 ± 0.08 ^de^	1.87 ± 0.08 ^e^	2.82 ± 0.17 ^cd^	3.96 ± 0.17 ^ab^	3.41 ± 0.28 ^bc^	4.68 ± 0.35 ^a^	2.76 ± 0.08 ^cd^	<0.001
Propionic	4.66 ± 0.26 ^c^	3.24 ± 0.16 ^c^	4.30 ± 0.35 ^c^	6.31 ± 0.48 ^b^	3.90 ± 0.12 ^c^	7.86 ± 0.28 ^ab^	8.96 ± 0.56 ^a^	6.75 ± 0.20 ^b^	<0.001
Lactic	44.1 ± 1.65 ^ab^	46.7 ± 1.21 ^a^	33.8 ± 2.97 ^bc^	33.4 ± 1.35 ^bc^	29.4 ± 3.83 ^c^	38.3 ± 0.94 ^abc^	32.5 ± 0.55 ^bc^	39.8 ± 4.60 ^abc^	<0.001
Succinic	5.04 ± 0.15 ^a^	5.43 ± 0.13 ^a^	2.11 ± 0.10 ^c^	2.47 ± 0.25 ^c^	3.68 ± 0.21 ^b^	2.75 ± 0.083 ^b^	0.45 ± 0.04 ^d^	0.91 ± 0.12 ^d^	<0.001

^1^ Values are means ± SEM, n = 3 fermentations per substrate. A one-way ANOVA model was used to assess the effect of substrate on all organic acids. Means in a column (i.e., substrate effect) with different letters differ (*p* ≤ 0.05). The valeric, iso-butyric, iso-valeric, heptanoic and hexanoic acid productions were negligible in the samples (i.e., below the detection limit) and, therefore, not reported. ND, not detected (i.e., below the detection limit).

**Table 3 microorganisms-13-01870-t003:** The total and individual organic acid productions after 18 h of in vitro fermentation per type of pip fruit using a pooled human faecal inoculum ^1^.

Organic Acid	Substrate			*p* Value
	Apple	Pear	Inulin	
	µmol/mL fermenta			
*N*	15	6	3	
Total ^2^	102 ± 1.48	106 ± 2.34	103 ± 3.31	0.360
Formic acid	0.90 ± 0.05 ^b^	2.23 ± 0.07 ^a^	2.32 ± 0.12 ^a^	<0.001
Acetic acid	52.7 ± 0.45	52.7 ± 0.71	51.3 ± 1.00	0.437
Butyric acid	2.47 ± 0.08 ^b^	4.05 ± 0.13 ^a^	2.76 ± 0.19 ^b^	<0.001
Propionic acid	4.48 ± 0.15 ^c^	8.41 ± 0.24 ^a^	6.75 ± 0.30 ^b^	<0.001
Lactic acid	37.5 ± 1.14	35.4 ± 1.80	39.8 ± 2.55	0.360
Succinic acid	3.75 ± 0.07 ^a^	1.60 ± 0.14 ^b^	0.91 ± 0.16 ^c^	<0.001

^1^ Values are means ± SEM. A one way ANOVA model was used to assess the effect of substrate for all organic acids. Means in a column (i.e., substrate effect) with different letters differ (*p* ≤ 0.05). The valeric, iso-butyric, iso-valeric, heptanoic, and hexanoic acid productions were negligible in the samples (i.e., below the detection limit) and, therefore, not reported. ^2^ The total organic acid production was defined as the sum of formic, acetic, propionic, butyric, lactic, and succinic acid concentrations.

## Data Availability

The 16S rRNA gene sequencing data files have been deposited into the Sequence Read Archive with links to BioProject accession number PRJNA1292375 in the NCBI BioProject database (https://www.ncbi.nlm.nih.gov/bioproject/ (accessed 17 July 2025)).

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
