# Peer review of "The Effect of Apple and Pear Cultivars on In Vitro Fermentation with Human Faecal Microbiota"

_microorganisms, 2025, doi:10.3390/microorganisms13081870_

Round 1

Reviewer 1 Report

Comments and Suggestions for Authors

Dear editors,

This manuscript presents a well-structured and logically coherent study exploring the effects of different apple and pear cultivars in vitro fermentation model. The experimental design is robust, and the results are generally clear and well-interpreted. However, some of the results described in the figures disappeared. Materials and Methods section should be clearer structuring.

1.The Introduction section provides clear logic and sufficient background on apples and pears, including their nutrients and bioactive components; however, more examples of related in vitro experiments should be included.

2.The Materials and Methods section needs clearer organization, e.g., subsections such as 2.1 Materials, 2.2 In vitro experiments, …, 2.n Data analysis.

3.Results regarding gut microbiota composition are missing, though partly discussed later; please clearly include these results related to figures. Also, diversity analysis is usually presented with α-diversity before β-diversity, please consider following the conventional order.

Author Response

"The Introduction section provides clear logic and sufficient background on apples and pears, including their nutrients and bioactive components; however, more examples of related in vitro experiments should be included."

  1. Thank you for this suggestion. We have included examples of related in vitro experiments in the Introduction. 

"The Materials and Methods section needs clearer organization, e.g., subsections such as 2.1 Materials, 2.2 In vitro experiments, …, 2.n Data analysis."

2. The Materials and Methods section has the subsections headings now. These signposts do make the section clearer.

"Results regarding gut microbiota composition are missing, though partly discussed later; please clearly include these results related to figures. Also, diversity analysis is usually presented with α-diversity before β-diversity, please consider following the conventional order."

3. This section has been reinstated. It appears to have been lost during formatting. And the diversity analyses are now presented in the conventional order. 

Reviewer 2 Report

Comments and Suggestions for Authors

The research investigates the effects of different apple and pear varieties, with varying polyphenol content, on the human gut microbiota, using an in vitro digestion and fecal fermentation model. Five apple and two pear varieties were analyzed, monitoring changes in pH, microbial composition, and organic acid production after 18 hours of fermentation. The results show that the varieties significantly influence the structure of the microbiota and the production of butyrate – an acid beneficial for gut health. For example, the “Golden Hornet” apple showed the highest microbial diversity, and the “PremP009” pear generated the highest butyrate production.
This study is notable for being among the few that analyzes the effects of apples and pears on the intestinal microbiota at the variety level, highlighting the influence of polyphenol differences on microbial composition and metabolism, using an in vitro model with rigorous experimental control, correlating fruit varieties with specific changes in the microbiota (such as the increase in the Acidaminococcus intestini species) and opening new directions for personalized nutrition and the use of distinct varieties in prevention or diet therapy. fruit varieties with precise changes in the microbiota (e.g.: the increase in the Acidaminococcus intestini species).

Author Response

Dear reviewer, thank you for your review.

Reviewer 3 Report

Comments and Suggestions for Authors

In the present study, the authors examined the effect of apple and pear cultivars on human gut microbiota and organic acids production. Experiments showed that apples and pear cultivars differently affect gut microbiota and metabolites. Honestly, the manuscript does not meet the general format. There is repetition in the Materials and Methods. Especially, the authors did not describe the results (after Figure 3) in text. It is difficult to understand the results in this manuscript. I strongly recommend that the authors should describe the results in the manuscript and re-submit the manuscript. I have some additional comments as follows:

1.    Please describe what is unknown to solve this manuscript in the Abstract and Introduction.  It is difficult to understand why the authors focus 5 apple and two pear cultivars and what problems or question do the authors want to solve. 
2.    How to determine the weight of fruit as 36g?
3.    Please describe validation of the simulation the small intestine absorbs small metabolites less than 10 kDa.
4.    How to determine the final concentration was 25g/100ml ?
5.    Please compare results pre and post fermentation pH and organic acid concentration.

Author Response

"Please describe what is unknown to solve this manuscript in the Abstract and Introduction.  It is difficult to understand why the authors focus 5 apple and two pear cultivars and what problems or question do the authors want to solve."

Thank you for helping us clarify the study purpose and science questions. We were interested in the effects on the structure and function of the gut microbiota between apple and pear fruits as well as traditional and new cultivars of these fruits grown under New Zealand conditions. Apples and pears both belong to the same family Rosaceae but have differences in fiber content, fruit shape, texture and flavor. Pears have fewer calories and lower fiber levels than apples.

"How to determine the weight of fruit as 36g?" 

The amount of material that was used for this study was based on previous experience in within the team and advice from our colleague John Monro who has extensive experience in in vitro digestion methodology. The size of the digestion vessel was also taken into account. In a similar study, Tenore et al. 2013 used 20 g of the test materials.

"Please describe validation of the simulation the small intestine absorbs small metabolites less than 10 kDa."

This size was selected to allow the polyphenols to be retained while allowing the sugars and other small molecules to be removed following the gastric digestion phase (lines 109-111).

"How to determine the final concentration was 25g/100ml ?"

This was determined from the substrate concentration before and after centrifugation of the digested substrate. The concentration was adjusted using the digested water blanks that were carried out alongside the apple and pear substrates.

"Please compare results pre and post fermentation pH and organic acid concentration."

That is an interesting suggestion. Unfortunately, we did not measure the pH or organic acid concentrations pre fermentation only post fermentation. This was because our focus was on the fermentation outcome between the different fruit. Thank you for this suggestion, we will endeavor to include this in future studies. 

Round 2

Reviewer 1 Report

Comments and Suggestions for Authors

This study aimed to investigate the effects of New Zealand grown apples and pears with differing polyphenol contents on the structure and function of the human gut microbiota. Five apple and two pear cultivars underwent in vitro human digestion and microbial fermentation. he impacts of apple and pear cultivars on gut microbial community structure and fermentation metabolites vary significantly in vitro. This study contributes to the understanding of the nutrient composition of different pip fruit cultivars that may contribute to the shaping of gut microbiota and promote the production of beneficial organic acids. However, the mechanisms of why different of fruit cultivars significantly shaped of gut microbiota and promoted the production of beneficial organic acids. Therefore, the discussion needs to delve deeper to uncover its underlying mechanisms.

Line 22, (could we include this analysis?).

Author Response

This study aimed to investigate the effects of New Zealand grown apples and pears with differing polyphenol contents on the structure and function of the human gut microbiota. Five apple and two pear cultivars underwent in vitro human digestion and microbial fermentation. he impacts of apple and pear cultivars on gut microbial community structure and fermentation metabolites vary significantly in vitro. This study contributes to the understanding of the nutrient composition of different pip fruit cultivars that may contribute to the shaping of gut microbiota and promote the production of beneficial organic acids. However, the mechanisms of why different of fruit cultivars significantly shaped of gut microbiota and promoted the production of beneficial organic acids. Therefore, the discussion needs to delve deeper to uncover its underlying mechanisms.

Line 22, (could we include this analysis?).

structure and function of the human gut microbiota. Five apple and two pear cultivars underwent in vitro human digestion and microbial fermentation. he impacts of apple and pear cultivars on gut microbial community structure and fermentation metabolites vary significantly in vitro. This study contributes to the understanding of the nutrient composition of different pip fruit cultivars that may contribute to the shaping of gut microbiota and promote the production of beneficial organic acids. However, the mechanisms of why different of fruit cultivars significantly shaped of gut microbiota and promoted the production of beneficial organic acids. Therefore, the discussion needs to delve deeper to uncover its underlying mechanisms.

Line 22, (could we include this analysis?).

Yes, the analysis is now in the manuscript. It was inadvertently deleted while moving figures around within the previous version of the manuscript.